**Cite this article:** van der Boor C, Andersen LS, Massazza A, Tol WA, Taban D, Roberts B, Ssebunnya J, Kinyanda E, May C, Nadkarni A and Fuhr D (2024). Using theory of change to plan for the implementation of a psychological intervention addressing alcohol use disorder and psychological distress in Uganda. *Cambridge Prisms: Global Mental Health*, **11**, e6, 1–11 https://doi.org/10.1017/gmh.2023.93

theory of change; brief psychological interventions; alcohol use disorders; mental health; conflict-affected populations

**Corresponding author:**
Catharina van der Boor;
Email: Catharina.Van-der-Boor@lshtm.ac.uk

# Using theory of change to plan for the implementation of a psychological intervention addressing alcohol use disorder and psychological distress in Uganda

Catharina van der Boor[1] ⓘ, Lena S. Andersen[2] ⓘ, Alessandro Massazza[1], Wietse A. Tol[2,3], Dalili Taban[4], Bayard Roberts[1] ⓘ, Joshua Ssebunnya[5], Eugene Kinyanda[5], Carl May[1], Abhijit Nadkarni[6,9] ⓘ and Daniela Fuhr[1,7,8] ⓘ

[1]Department of Health Services Research and Policy, London School of Hygiene and Tropical Medicine, London, UK; [2]Section of Global Health, Department of Public Health, University of Copenhagen, Copenhagen, Denmark; [3]Athena Research Institute, Vrije Universiteit Amsterdam, Amsterdam, The Netherlands; [4]HealthRight International, Kampala, Uganda; [5]Mental Health Focus Area, MRC/UVRI & LSHTM Uganda Research Unit/MRC Investigator, Entebbe, Uganda; [6]Department of Population Health, London School of Hygiene and Tropical Medicine, London, UK; [7]Department of Prevention and Evaluation, Leibniz Institute for Prevention Research and Epidemiology, Bremen, Germany; [8]Health Sciences, University of Bremen, Bremen, Germany and [9]Addictions and Related Research Group, Sangath, Goa, India

## Abstract

In conflict-affected settings, prevalence of alcohol use disorders (AUDs) can be high. However, limited practical information exists on AUD management in low-income settings. Using a theory of change (ToC) approach, we aimed to identify pathways influencing the implementation and maintenance of a new transdiagnostic psychological intervention ("CHANGE"), targeting both psychological distress and AUDs in humanitarian settings. Three half-day workshops in Uganda engaged 41 stakeholders to develop a ToC map. ToC is a participatory program theory approach aiming to create a visual representation of how and why an intervention leads to specific outcomes. Additionally, five semi-structured interviews were conducted to explore experiences of stakeholders that participated in the ToC workshops. Two necessary pathways influencing the implementation and maintenance of CHANGE were identified: *policy impact,* and *mental health service delivery.* Barriers identified included policy gaps, limited recognition of social determinants and the need for integrated follow-up care. Interviewed participants valued ToC's participatory approach and expressed concerns about its adaptability in continuously changing contexts (e.g., humanitarian settings). Our study underscores ToC's value in delineating context-specific outcomes and identifies areas requiring further attention. It emphasizes the importance of early planning and stakeholder engagement for sustainable implementation of psychological interventions in humanitarian settings.

## Impact statement

By using theory of change (ToC), we use a participatory approach to understand different pathways that can shape the implementation and maintenance of a new psychological intervention named "CHANGE" in a humanitarian setting in Uganda. The CHANGE intervention is designed to address both psychological distress and alcohol use disorders in conflict affected populations. The findings of this study reveal that there are two pathways that are important for the implementation of CHANGE in this context: policy impact, and mental health service delivery. We reflect on the utility of using a ToC methodology in this context and identify areas that require further attention.

## Background

Over the last 10 years, the number of forcibly displaced populations resulting from armed conflict, persecution and/or violence has grown by more than 50% (UNHCR, 2021). In 2021 alone, 23 countries faced high or medium intensity conflicts (World Bank, 2022), which have far-reaching social, economic and health impacts (Milián et al., 2022; World Bank, 2022).

Armed conflict can impact mental health both directly, for example, by exposing people to traumatic events, and indirectly, for example, by worsening known social determinants of mental health (i.e., impoverishment, poor access to healthcare, loss of social networks) (Lo et al., 2017a). In these contexts, alcohol use disorders (AUDs) may provide unhealthy ways to cope with

chronic stressors, other psychological consequences of conflict exposure, and can further compound risk factors (Weaver and Roberts, 2010; Ezard, 2012; Lo et al., 2017a).

Literature on AUDs demonstrates an increased risk for mental health disorders (Grant et al., 2004; Lo et al., 2017a), noncommunicable diseases (Hammer et al., 2018; Nadkarni et al., 2023a), gender-based violence (Shiva et al., 2021) and wider social and economic risks (Hammer et al., 2018). People with AUDs often experience barriers to treatment which can reduce health seeking behavior and underutilization of services (Probst et al., 2015; Mellinger et al., 2018; McCrady et al., 2020).

Addressing AUDs in humanitarian settings is included in international humanitarian response guidelines (IASC, 2007; Sphere, 2018). However, there remains limited evidence to guide the management of AUDs among conflict-affected people in low-income settings (Roberts and Ezard, 2015; Kane and Greene, 2018; Fuhr et al., 2021). The CHANGE consortium aims to address this evidence gap, through the development and implementation of a new psychological intervention ("CHANGE") addressing both AUDs and psychological distress among conflict-affected men (https://www.lshtm.ac.uk /change; Fuhr et al., 2021). The CHANGE intervention is informed by Problem Management Plus (PM+) (Dawson et al., 2015). PM+ is a transdiagnostic (i.e., noncondition specific) intervention developed by the WHO, that improves symptoms of depression, anxiety and psychological distress (Schäfer et al., 2023). A transdiagnostic approach can be useful as most people present with comorbidity, while simultaneously making it more feasible to deliver in resource-poor settings (Dawson et al., 2015). PM+ includes strategies based on cognitive behavioral therapy, including problem solving, stress management, behavioral activation and access to social support. Global evidence has demonstrated the efficacy of PM+ at reducing psychological distress in low and middle-income settings (Schäfer et al., 2023). However, PM+ does not include strategies for AUDs which is a major but neglected issue among conflict-affected populations (Roberts et al., 2011; Kane and Greene, 2018). The CHANGE intervention seeks to address this gap, by including strategies into PM+ that can target AUDs (such as personal feedback, goal setting and planning change, cognitive behavioral skills such as drink refusal, and relapse management). CHANGE is delivered by lay health workers and is composed of six individual face-to-face sessions (Fuhr et al., 2021; Nadkarni et al., 2023b). The effectiveness of CHANGE is currently being evaluated through a randomized controlled trial with South Sudanese refugee men in Rhino camp settlement, northern Uganda.

When developing complex interventions, it is recommended to investigate how context may influence intervention outcomes and implementation (Craig et al., 2008; De Silva et al., 2014).

Theory of change (ToC) is a participatory planning process focused on how a project can achieve long-term outcomes through a logical sequence of intermediate outcomes (Vogel, 2012). It enables systematic identification of knowledge gaps in context and can provide a comprehensive set of indicators to evaluate the stages of causal pathways through which complex interventions may achieve impact (De Silva et al., 2014). It is visually represented in a ToC map, which shows the ways in which an intervention is anticipated to produce its effects within the constraints of the specific environment in which it is put into practice (De Silva et al., 2014).

The aim of this study was to use ToC to summarize pathways to the implementation and maintenance of the CHANGE intervention for refugees in the context of a refugee settlement in northern Uganda. It was also developed to investigate barriers and facilitators toward implementation and explore strategies that may overcome certain barriers. The specific objectives of this study were to (a) explore context-specific intermediate and long-term outcomes necessary for sustainable implementation of CHANGE, including policy and health service considerations; (b) identify barriers and facilitators underlying each outcome and (c) reflect on the conduct of the ToC process through stakeholder interviews. To the best of our knowledge, this article is the first to report on the use of ToC to shed light on pathways for a public health intervention for AUDs among refugees.

## Methods

### Study setting

Data collection occurred between June 2021 and February 2022 in Kampala, Arua and Rhino camp settlement in Uganda. An estimated 136,900 refugees currently live in Rhino camp settlement in impoverished conditions (UNHCR, 2019c). Arua is the largest urban area near the settlement, where many humanitarian agencies operating in the settlement have their offices. Kampala, Arua and Rhino camp settlement were chosen to have a varied sample of experienced stakeholders working with refugees.

Uganda's Ministry of Health has included mental health care into its general health provision, outlined in the national Health Sector Integrated Refugee Response Plan (Uganda Ministry of Health, 2019). However, there is only one mention of the need to prevent and control noncommunicable diseases including substance use disorder, with no further detail or guidelines on how this can be achieved. Those living in the settlement have the right to access free healthcare through 13 primary care posts distributed across the settlement (UNHCR, 2019a, 2019b). However, according to a needs and resource assessment conducted in Rhino camp in 2016, accessibility to general healthcare was limited, and there were very limited mental health and psychosocial support services (MHPSS) available (Adaku et al., 2016). As such, despite efforts, there are still gaps in responding to the MHPSS needs of refugee communities, including poorly equipped health centers, and shortages of healthcare workers and medication (Baingana and Patrick, 2011; Rokhideh, 2017; Kane and Greene, 2018). The United Nations High Commissioner for Refugees (UNHCR) has highlighted the need for scalable community-led interventions to bridge the needs gap (Kaltenbacher, 2019).

### Participatory theory of change workshops

#### Theory of change

Three half-day face-to-face workshops gathered information for the development of a ToC map. ToC is often developed using a backward mapping process, starting with the long-term outcomes and then mapping the required process of change. Achieving long-term outcomes depends on meeting a series of intermediate outcomes. Rationales justify these intermediate outcomes, and explain how specific interventions (activities) will contribute to their achievement. Assumptions (i.e., barriers/facilitators) required for the ToC are also recorded. Ultimately, ToC allows for multiple causal pathways, interventions and feedback loops to achieve specific outcomes, while also acknowledging the role of context for implementation (Vogel, 2012; Breuer et al., 2015).

### Participants and recruitment

Stakeholders involved with refugee services and/or with MHPSS programming with South Sudanese refugees in Uganda were

recruited. The final sample included national stakeholders working for the government (N = 3), national and international NGO workers (N = 14) and other community health and social care workers (N = 24). Workshop 1 was conducted in Kampala (N = 15), workshop 2 in Arua (N = 13) and workshop 3 in Rhino camp settlement (N = 12) (see Table 1 for an overview). All participants provided informed consent.

## Procedure of workshops

The ToC study was carried out during the formative research phase, that is, prior to the feasibility and effectiveness trials of the CHANGE intervention. In Kampala, the ToC workshop was facilitated by a Ugandan clinical psychologist and researcher with experience of ToC. In Arua and the Rhino camp settlement, they were facilitated by the CHANGE project coordinator, who had received

**Table 1.** Description of ToC participants

| ToC attended | Gender | Job title |
|---|---|---|
| Arua | M | Community services officer |
| | F | Registered mental health officer at NGO |
| | M | MEAL Coordinator at NGO |
| | F | Assistant chief administrative officer for the local government |
| | F | Clinical Psychologist at NGO |
| | F | Psychiatric Nurse at NGO |
| | M | Health Tutor at local nursing school |
| | F | Lecturer at local university |
| | M | Clinical Officer |
| | M | Consultant |
| | F | Registered Mental Health Nurse |
| | M | Psychiatric clinical officer at regional hospital |
| | M | Psychologist at NGO |
| Rhino refugee settlement | M | Refugee block leader within one of the villages of Rhino camp settlement, reports to the Refugee Welfare Council |
| | M | In charge of the resources for the local health center |
| | F | Secretary general for persons with special needs |
| | M | Refugee opinion leader, involved in important decisions concerning refugee wellbeing |
| | M | Refugee Leader who works in the Refugee Welfare Council to ensure wellbeing within the community |
| | F | Refugee Women Representative, responsible for matters concerning refugee women in the village |
| | M | Refugee Welfare Council chairperson, oversees wellbeing and provides leadership including making decisions on matters concerning refugees |
| | M | Refugee Religious leader, provides spiritual guidance and counseling |
| | M | Refugee Pastor, provides spiritual guidance and counseling |
| | F | Refugee welfare council chairperson, oversees wellbeing and provides leadership including making decisions on matters concerning refugees |
| | M | Church Leader, provides spiritual guidance and counseling |
| | M | Refugee secretary general, responsible for person with special needs |
| | M | Refugee Church leader, provides spiritual guidance and counseling |
| Kampala | M | Project coordinator for NGO |
| | M | Consultant at national hospital |
| | F | MHPSS supervisor at NGO |
| | F | Acting associate commissioner mental health |
| | M | Senior consultant psychiatrist at national hospital |
| | M | Head of department of mental health at national university |
| | F | Clinical Psychologist |
| | F | Mental health team leader at NGO |
| | M | Clinical specialist at NGO |
| | M | Country Director at NGO |

(*Continued*)

**Table 1.** (*Continued*)

| ToC attended | Gender | Job title |
|---|---|---|
| | F | Program associate |
| | M | Program Director at NGO |
| | F | Program officer |
| | M | M&E Coordinator at NGO |
| | M | Psychiatry Doctor |

previous training. Workshops were in English and audio recorded. At the start of the workshops, participants were given an overview of the main components of ToC, and discussions were then held around each of these main components. Post-it notes were used to iteratively develop the ToC map. Participants of the workshops led the discussions and were guided by two facilitators. Once the three workshops had taken place, researchers LSA and DF compared the different components that had emerged in each of the three workshops and drafted one final ToC map that reflected the views of the three groups. Audio recordings were used to check missing information and details when necessary. After creating the draft map, 16 participants provided written feedback on the draft map via email, which was reviewed and included in the final ToC map.

### Semi-structured interviews

The workshops were followed by individual semi-structured interviews with a subset ($n = 5$) of stakeholders who participated in the ToC workshops. The interview aim was to explore the experience of workshop participation, and views on the strengths and limitations of ToC. Participants were purposefully sampled based on time and availability.

### Procedures for interviews

Semi-structured interviews took place via zoom with researcher AM. The topic guide covered the following domains: defining ToC in their own words, perceived purpose of ToC, role of facilitator, participant interactions, differing views and strengths and limitations (see additional files). Following the first interview, the topic guide was edited and remained unchanged for the rest of the data collection. Interviews lasted between 30 min and 1 h. All interviews were in English, audio recorded and transcribed *verbatim*.

Thematic analysis was used (Braun and Clarke, 2022). As the data were transcribed, recurring themes were noted that informed the preliminary coding framework. A largely inductive coding approach was used by identifying codes that were most frequently represented in the dataset. The preliminary coding framework was discussed among the core members of the team and then finalized. All data were coded and analyzed in NVivo 12.

## Results

### ToC for the CHANGE intervention

Figure 1 presents the ToC map. Two distinct causal pathways were identified that could facilitate implementation of the CHANGE intervention at different levels of the health system: (a) *Policy pathway*, and (b) *Mental health service delivery pathway*. Eight interventions (interventions 1–8) and 12 assumptions (assumptions A–L) were identified by stakeholders (Figure 1).

### Pathways to improved health and wellbeing

The two pathways (*Policy impact*, and *Mental health service delivery*) that lead to the overall impact, that is, improved health and wellbeing, are associated with four long-term outcomes: (a) a reduction in AUDs and associated mental health disorders in refugee communities; (b) improved social and economic productivity; (c) improved family stability and functioning and (d) reduction in perpetration of gender-based violence resulting from AUDs (Figure 1). The two pathways are described below.

### Policy impact pathway

The first pathway to support the maintenance of the CHANGE intervention is policy-focused. Assuming the intervention's effectiveness, there is a need for policies to be in place that continue to prioritize and promote the mental health and wellbeing of refugee populations. To achieve this, political leaders must demonstrate a commitment to prioritizing mental health care (assumption I). Supportive environments can aid help seeking, and promote mental health, among others. An early intermediate outcome of such supportive policies is the availability of political leadership that is sensitized to the need to address AUDs in refugee populations both at a national and local level, which can facilitate the development and implementation of a mental health plan targeting AUDs in refugee populations.

Workshop discussions extensively covered the reciprocal relationship between AUDs and social determinants of health (e.g., employment). Stakeholders recognized that treating AUDs alone would not suffice to improve mental health and wellbeing. Political leaders must recognize this reciprocal relationship (assumption J). Stakeholders noted the negative impact of high unemployment levels within refugee populations in Uganda and highlighted the need for clients with AUDs to be supported to earn an income. Three key interventions were proposed (a) establishing associations to promote income generating activities, that is, training households in small business set up and/or providing small cash grants (intervention 6); (b) introducing cash-for-work programs, that is, building infrastructure, garbage collection (intervention 7) and (c) recognizing farming as a viable business for refugees (intervention 8). These depend on the assumption that those who recover from mental health disorders and/or AUDs, and receive vocational or skill-based trainings, can then actively participate in various programs and activities (assumption H).

Moreover, attention is required for the relationship between AUDs and social determinants within both national mental health and refugee response plans. This involved recognizing crucial social determinants, like unemployment or housing status, that can increase the risk of AUDs. This, in turn, demands the allocation of resources and funding. Resources should be channeled toward enhancing the mental health plan to include strategies addressing

substance misuse (assumption K), contingent on political will and priorities. Funding should also be directed toward the development and maintenance of interventions, including the three key interventions outlined above, and task-sharing interventions like CHANGE, that can address specific community challenges such as mental health disorders and AUDs (assumption L).

### Mental health service delivery

The second pathway relates to mental health service delivery, including the identification of individuals with AUDs and the delivery of targeted community-based interventions. An early intermediate outcome of this service delivery is growing awareness within the community regarding AUDs which can promote support and sensitize community members toward the harmful effects of alcohol.

To bolster this awareness and sensitization surrounding AUDs and available treatments, involving Refugee Welfare Councils and incorporating messaging on AUDs within community-based religious settings (intervention 1) may be essential. Furthermore, this awareness may facilitate the identification of refugees who would benefit from support. It was further speculated that the identification of individuals with AUDs and support in accessing care hinges on a peaceful and stable environment (assumption A). Additionally, it was emphasized that community health workers need training in screening and identification of people with AUDs, and in the selection of appropriate referral pathways (intervention 2). This assumes the existence of suitable and identifiable referral pathways (assumption C), and that refugees with AUDs are willing to seek help (intervention 1). The ability to seek help may also be contingent on the absence of infectious disease outbreaks in the community (assumption B).

Those who seek help and are screened and identified as having severe alcohol dependence and/or are at risk of suicide should be referred for specialized treatment, where available. Those with less severe AUDs can receive interventions from lay workers in community settings. Community workers with a healthcare background (i.e., social workers and counselors) should be trained to administer these interventions (i.e., CHANGE). While training local workers may improve case management within the health system (intervention 3), it is dependent on the availability of human resources (assumption E).

Once the CHANGE intervention has been completed, refugees can receive follow-up care in the community. This may include vocational skills training (intervention 4) and establishing support groups to foster social support and aid ongoing recovery and/or symptom reduction (intervention 5). There was recognition that for follow-up care to be successful, individuals' support networks (e.g., family members and others) must also contribute to the process by encouraging engagement (assumption F). Additionally, there needs to be availability of human and non-human resources to provide vocational trainings, and other discussed programs and activities (assumption G).

### Experiences with ToC

Five workshop participants were interviewed, of which four were male and one was female. Three worked for an international NGO, one was a clinical officer in Rhino camp settlement, and one was a psychiatric nurse in the Arua regional hospital. Various themes were discussed, which are summarized below under three headings: (a) ToC definition and implementation, (b) advantages and disadvantages of ToC and (c) areas of future improvement.

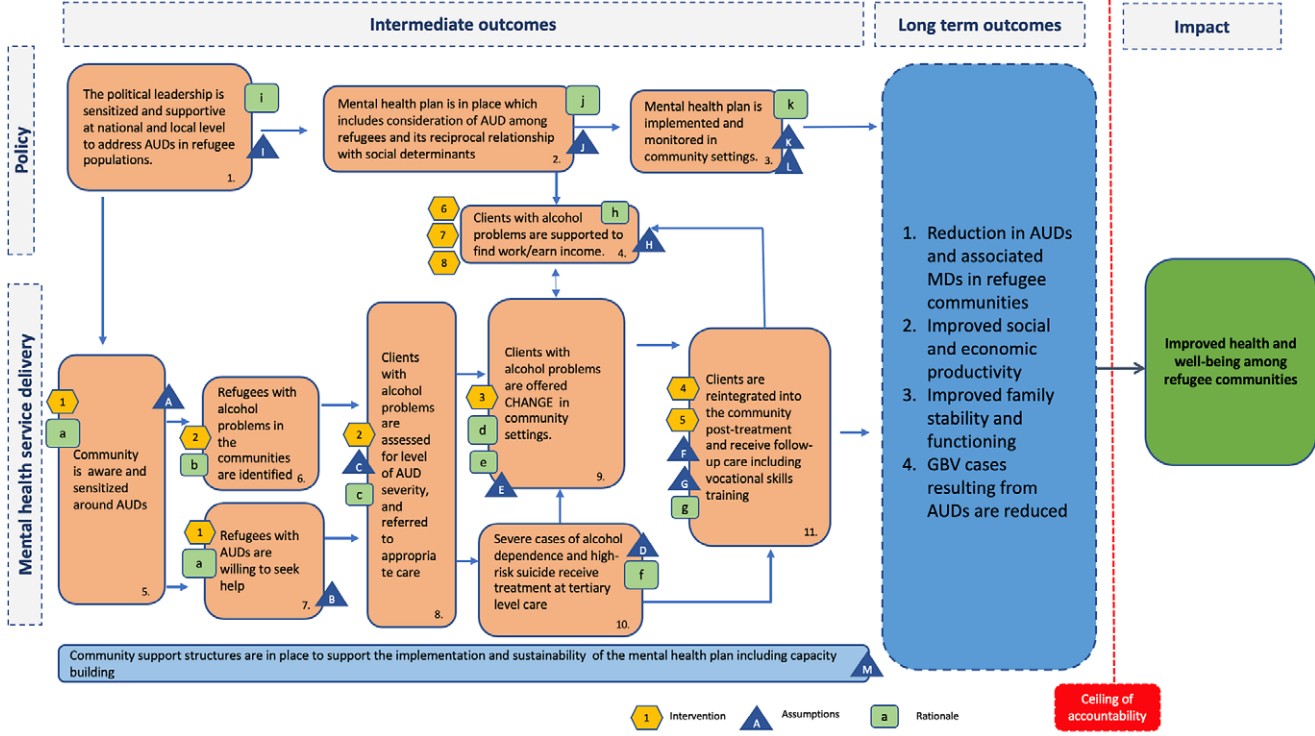

*Outcome 9 is within the scope of the CHANGE intervention

**Figure 1.** (Continued)

| Interventions | 1 |
|---|---|
| 1) Provide awareness and sensitization around AUDs and treatment availability in the community including incorporating AUD messages within church preaching and involving Refugee Welfare Councils operating within the settlement. | |
| 2) Train community health workers to screen for and identify AUDs in the refugee community and to select appropriate referral pathways (based on levels of severity and other associated problems). | |
| 3) Train social workers and counselors in administering CHANGE and improve capacity for better case management. | |
| 4) Provide refugees in recovery from AUDs with vocational skills training. | |
| 5) Establish community support groups to aid in maintaining reduction in alcohol use or recovery. | |
| 6) Form association for income generating activities to ensure refugees can engage in meaningful activities. | |
| 7) Introduce cash for work in the settlement to reduce unemployment. | |
| 8) Introduce farming as a business for refugees. | |

| Assumptions | A |
|---|---|
| A. There is peace and stability in the refugee settlements. | |
| B. There is no total lockdown due to infectious disease outbreaks. | |
| C. Referral pathways are identified. | |
| D. Mental health services are accessible to refugees. | |
| E. Health care workers including social workers, community health care workers, and counselors are available to provide services locally. | |
| F. Support network is willing to support follow up care. | |
| G. Resources are made available for the provision of vocational skills training. | |
| H. Refugees who receive vocational skills training find employment after reduction in alcohol use or recovery. | |
| I. Political leaders are willing to prioritize mental health care. | |
| J. Political leaders recognize the association between mental health and social determinants. | |
| K. Resources and funding are made available for the mental health plan. | |
| L. Mental health plan supports the implementation of CHANGE. | |
| M. Community support structures are in place to support the implementation and sustainability of the mental health plan including capacity building. | |

| Rationale | a |
|---|---|
| a) Reducing stigma towards people with an alcohol use disorder can help facilitate the successful. implementation of substance misuse interventions in conflict-affected populations (Greene et al., 2018). | |
| b) Community case finding can be an effective strategy for detecting people with mental disorders and linking them to mental health care (Jordans et al., 2019). | |
| c) Early detection and management of alcohol use disorders is key to reducing problems related to alcohol use (38). | |
| d) Community based interventions for substance use disorders are recommended by the National Institute for Health and Care Excellent (39). | |
| e) Task-shifting mental health treatment delivery to non-specialist providers in low resource settings is recommended by the World Health Organization (40). | |
| f) Alcohol withdrawal syndrome is potentially life-threatening and requires medical management (41). | |
| g) Follow up care for individuals with alcohol use disorders improves long-term outcomes (42). | |
| h) The bidirectional relationship between certain poverty dimensions and common mental disorders has been established (43). | |
| i) The Ugandan government acknowledges the potential "adverse health, social and economic consequences" of alcohol in their national alcohol control policy (30). | |
| j) The social and cultural environment influences alcohol use and misuse (44). | |
| k) The WHO recommends integrating mental health services into community and primary health care settings to address the global mental health treatment gap (45). | |

**Figure 1.** Theory of change map.

### ToC definition and implementation

When asked to provide a definition for ToC, and what it was about, participants provided heterogeneous responses (see Table 2). A common focus was on the determination of a causal pathway whereby the ToC worked as a way of mapping outcomes necessary for a certain impact to take place, "what needs to be done in order to achieve that impact." One participant described ToCs as a way of giving direction, and another as that of ensuring that the intervention "flows logically." The ToC notion of "backward mapping" (i.e., starting from impact and going backward to the earliest outcome that must be realized) was perceived as particularly useful for this.

> A theory of change, is, I would call it, a process of… evolving, a process of developing an idea from, the end to the beginning, […] let me give an example, here, you are going, you have a journey to make and this journey from UK to Uganda, […] it's like you reverse, […] what steps do I need to take before I board in Uganda, what activities should I do before I arrive in Uganda, what objective do I have to achieve, so like you are going in reverse in a theory of change (Hospital psychiatric nurse, Arua).

Respondents highlighted the value of having different types of knowledge and experiences represented. For example, some mentioned the importance of avoiding mixing participants with very different types of status (e.g., mixing refugees with policy makers) out of fear that some may feel inhibited from speaking out. Not everyone agreed with this, and one participant highlighted how it would be important to always have the direct end-users (e.g., refugees with AUDs) to ground the perspective of the group.

**Table 2.** Definitions of ToC construct according to different participants

| Definition of ToCs for participants |
| --- |
| "Ehm… in my own words a ToC is a participatory, ehm, method where, ehm, all key stakeholders are involved in designing and the implementation and the subsequent running of an intervention to achieve specifically long-term, a specific impact… which clearly explains the causal pathways where this kind of, 'if you do this, it leads to this, if you do this to this, this to this,' which takes you to the final impact of the intervention…" (Project coordinator at NGO, Arua) |
| "I would say a ToC is, ehm, about understanding the impacts of an intervention, for example, if you are doing something what do you [?], the impact of this to be on the beneficiaries, the direct beneficiaries and the larger community, and, also what you need to do in order to achieve, in order to realize that impact on the beneficiaries and the larger community" (Community services officer at NGO, Arua) |
| "A theory of change is a like, in my own words, is the journey that one follows ehm… to achieve a goal, so you are setting out to move on a long journey, but you need to know what hurdles, what easy things that you can get on the way to reach your destination" (Clinical psychologist at NGO, Kampala) |
| "A theory of change, ehm… is, I would call it, a process of… ehm… evolving, a process of developing an idea from, ehm, the end to the beginning" (Hospital psychiatric nurse, Arua) |

We wanted the people to speak their mind, objectively, free, without fear of being underlook, you can imagine bringing a refugee all the way from the settlement and sitting in Kampala, in a five star hotel and being with the ministry people definitely he will underlook at his or her opinion in that setting, so we decided to ensure that have an environment which is fair to each of the stakeholder (Project coordinator at NGO, Arua).

The inclusion of the right participants and expertise was perceived as essential to reflect the reality on the ground.

I think one of the, one of the strengths is that, the, the ToC map that was produced is an outcome of critical reflection of people who are, who have been involved in the field who have sort of know the needs of the refugees in terms of mental health and psychosocial support, I feel that the ToC reflects what the real needs are and what can be done to address those needs (Community services officer at international NGO, Arua).

Two participants described that while the ultimate outcome of the ToC in the three workshops in Uganda was similar, that is, improved wellbeing for refugees, the pathways through which this outcome was reached differed across groups depending on the type of expertise and knowledge people brought.

The ToC notes for Rhino [refugee leaders as participants] you find that people, they talk more of interventions, individual interventions but it seems to have that gap of knowledge where to begin and where to get services, but if you look at Arua and Kampala [more policy makers and service providers involved] they look at putting the policies in place (Project coordinator at NGO, Arua).

The interaction between participants was described as overwhelmingly positive "the relationship was marvelous, people were friendly to each other, they were willing to learn." Where opinions differed, consensus was reached through discussion.

### Advantages and disadvantages of ToC
The ToC as a method was perceived to have multiple advantages. Clear advantages included empowerment, the visual and participatory nature of the workshops and a sense of ownership.

The room was filled with excitement, "oh these guys want us to develop our own intervention, this is so amazing!", and they had great remarks at the end of that, "if all interventions which many partners want to address problems in the community are done the way we did then", they say "they will have a lasting solution to their problems" (Project coordinator at NGO, Arua).

The main issue discussed was the ToC terminology, often perceived as confusing with participants reporting struggling to effectively differentiate between constructs (e.g., assumptions, interventions).

Understanding the different components was described as "tedious" and time consuming, and potentially problematic if constructs were not well understood.

The one in Rhino camp, was also based on experience of the refugees but I think it was a little bit challenging for them because some of these concepts appeared to be hard, [facilitator] help them to you know explain some of these things because it was the first time someone was coming across words like impacts or intermediate outcomes, objectives, outcomes, so… […] based on the low level of education of the refugees, of course mentioning impact that will be a very strange word to them and then they will ask "what do you mean by impact"? (Community services officer at NGO, Arua).

Participants also questioned the flexibility of ToC maps in volatile situations such as settings affected by armed conflict.

In an event where the situation on the ground changes, I mean the context in which the intervention was designed changes, I am trying to see whether, if the interventions do not achieve the intended purpose which was ideally planned, I don't know how that can fit in it, whether it has room for adjustments… (Project coordinator at NGO, Arua).

### Areas of future improvement
Participants mentioned it may have been helpful to have wider representation of certain stakeholders (e.g., primary healthcare workers, refugees with AUDs). To address the ambiguity around the ToC terminology, some participants suggested having appropriate simplifications, explanations of constructs and background materials before the meeting (e.g., material on the study and ToCs).

I think… one thing that, ehm, could be done differently to improve future ToCs is, […] describing in advance the ToC to the workshop participants so that they come into the workshop with an idea of how it will look (Hospital psychiatric nurse, Arua).

## Discussion

To the best of our knowledge, this article is the first to report on the use of ToC to support the planning of the sustainable implementation of a public health intervention to address AUDs among refugees. The findings in the current article highlight the utility of ToC to define and map context-specific outcomes across different implementation pathways and identify areas that require attention. Two necessary pathways were identified that will influence the implementation and maintenance of CHANGE in Uganda: a *policy impact pathway* and a *mental health service delivery* pathway. Both highlight the need for early planning and engagement of key

stakeholders to set the stage for implementation. The results of this study will help inform our subsequent work that focuses on examining the scalability of the CHANGE intervention through the health system and other humanitarian sectors.

### Policy impact pathway

The findings highlight steps required for the sustainable implementation of programs like CHANGE. First, within the policy impact pathway, it is necessary to establish government health policies that consider AUDs among refugees and acknowledge the role of social determinants (i.e., livelihoods, housing) on the development and persistence of AUDs. While Uganda's national alcohol control policy (Government of Uganda, 2019) recognizes the harmful impacts of alcohol, it falls short in acknowledging the role of social determinants on AUD development. Addressing these policy gaps is imperative for the long-term implementation and integration of interventions like CHANGE into community healthcare.

Second, the bidirectional relationship between specific dimensions of poverty and common mental health disorders and AUDs needs addressing. Research consistently demonstrates elevated risks for developing common mental health disorders and AUDs among individuals with limited access to material resources (Marmot et al., 2008; De Silva et al., 2011; Nadkarni et al., 2013; Allen et al., 2014; Lo et al., 2017b; Kibicho and Campbell, 2019). A previous study in Uganda with refugees showed high prevalence of post-traumatic stress, major depressive, generalized anxiety and substance use disorders, all linked to unmet basic needs (Bapolisi et al., 2020). Conversely, interventions addressing social determinants, for example, income-generating programs, can act as protective factors against initial AUD development (Amosu et al., 2016; Aaraj et al., 2021; Naseh et al., 2021), while potentially preventing relapse post-treatment. Failing to recognize the impact of social determinants on mental health and AUDs will impede efforts in prevention, recovery and symptom reduction, especially in resource-poor settings. For CHANGE, collaboration with other sectors addressing social determinants of health is necessary.

Third, relevant policies and national mental health plans need to be funded, implemented and closely monitored in the community. This includes integrating mental health services into community and primary healthcare settings to bridge existing treatment gaps (Jordans et al., 2019). Uganda's Ministry of Health has included mental health into general healthcare provisions, as outlined in the Uganda's Health Sector Integrated Refugee Response Plan of 2019 and the UNHCR' Public Health Strategic Plan for Uganda (Government of Uganda and UNHCR, 2022). The Refugee Response Plan states as a key priority the reinforcement of MHPSS services and infrastructure. However, limited progress has been made on this front leaving a big gap between the number of people who need this support and those who receive it. In 2019, it was reported that only 29% of refugees identified as needing MHPSS received it (Government of Uganda and UNHCR, 2022). The current primary healthcare facilities available to refugees are over-stretched and underfunded, and data are missing on the number of health facilities in settlements that have a mental health focal point (Government of Uganda and UNHCR, 2022). Task-shifting mental health treatment delivery to non-specialist providers, as recommended by the World Health Organization (WHO 2007), could enhance access to care, particularly through interventions by lay healthcare workers within the community. The 2019 national alcohol policy, issued by the Ministry of Health, also underscores the need for a coordinated approach to reduce harmful alcohol use

(Ministry of Health, 2019); however, the existing policy gap in addressing the social determinants of alcohol use, especially for vulnerable populations like refugees, persists.

### Mental health service delivery

Stakeholders emphasized the critical role of mental health service delivery in implementing and maintaining CHANGE. However, participants currently perceive there to be high levels of mental health stigma, which acts as a barrier to mental health care delivery. As such, delivering mental health care is dependent on community sensitization efforts of mental health and AUDs, enabling individuals to recognize the need for support, be informed about available resources, and reduce stigma. Nalwadda and colleagues previously assessed AUD prevalence among men in rural Uganda, along with levels of internalized stigma and attitudes toward seeking help (Nalwadda et al., 2018). Nearly half of those with AUDs reported that alcohol had "ruined their lives," yet none of the affected men had received treatment, and 55.5% believed AUDs to be untreatable. In another study on stakeholder perspectives in Uganda, mental health stigma was identified as a widespread issue, rooted in negative local beliefs. These included notions that people with mental health disorders are possessed by spirits, facing consequences for their actions, or that detoxification has serious side effects including intellectual disabilities (Ssebunnya et al., 2009). Such beliefs hinder the willingness and ability to seek help, potentially worsening existing mental health disorders, including AUDs, and perpetuating poverty cycles.

The current findings indicate that there is a perceived lack of awareness among stakeholders around tackling AUD in Rhino camp settlement. A prior study highlighted similar findings and that reducing stigma toward people with AUDs may facilitate the successful implementation of AUD interventions in conflict-affected populations (Greene et al., 2018). Participants in the current study proposed community-level resources to enhance awareness for AUDs, and available services to facilitate help-seeking. Several interventions, including educational initiatives, awareness campaigns and stigma reduction programs may prove helpful. This could include training community-based health workers in task-sharing interventions to support access to care. Importantly, support should extend beyond treatment completion, necessitating community-based follow-up care to aid reintegration and sustained wellbeing. Workshop participants highlighted the need for integrating psychosocial interventions into existing clinic-based treatment programs and creating opportunities for follow-up care to aid recovery and symptom reduction in the community.

Workshop participants called for the prioritization of further research on the impact of social determinants on mental health and AUDs in Rhino camp settlement. This knowledge can then be used to devise strategies to promote mental health and recovery within this context. For CHANGE, these findings highlight the importance of working with local community leaders to understand how these determinants may impact on AUD interventions and work together to promote the potential long-term benefits if the CHANGE intervention is proven to be effective. The benefits of using ToC workshops to develop a logical plan for scaling up mental health programs, contextualizing the programs and getting stakeholder buy-in has previously been highlighted in other resource-poor settings (Breuer et al., 2014; Fuhr et al., 2020; Babatunde et al., 2022). For the current project, the ToC map will need to be reviewed and adapted as further findings emerge on the implementation and scalability of CHANGE.

### Experiences of ToC

Beyond the ToC map, the current study explored participants' reflections on taking part in the workshops, through individual interviews. Participants recognized the benefits of the participatory nature of the workshops, specifically the sense of ownership over ideas. Participants did voice concerns over the flexibility of the final ToC map, as its validity may be challenged in settings where drastic changes can occur, such as humanitarian and/or conflict settings. One study (Breuer et al., 2014) previously described how to adapt the ToC methodology in order to ensure that it remains a useful method to use in shifting conditions (i.e., conflict-affected contexts). They suggested exploring "living ToCs" which can facilitate responsive programming throughout a program's life cycle. Living ToCs can include conducting regular program evaluations that allow for robust conclusions to be made on the impact of the program, which can then be fed back into the ToC and consequently lead to the adjustment of programing and desired outcomes.

### Strengths and limitations

The current research has facilitated the understanding of how the CHANGE intervention can impact change within Uganda's humanitarian context using a participatory approach. The decisions on how to implement and maintain the intervention need to be driven by local knowledge on operational dynamics, divergent interests, inconsistent funding and the need for safe access to those who can benefit from the intervention (Puri et al., 2015).

These workshops have enabled us to articulate clear outcomes and have helped draw lessons from the stakeholders' experience of working within the health sector both at a national and local level. The current article is the first article to use ToC for an AUD and mental health intervention in Uganda (and indeed with forcibly displaced persons globally).

This research had several limitations. First, the ToC map is based on the implementation of the CHANGE intervention in humanitarian settings in Uganda; however, the effectiveness of the CHANGE intervention still needs to be assessed. Second, stakeholders with lived experience (i.e., refugees) were only included in the ToC workshops when they held positions of responsibility within the settlement, meaning some insights regarding the on-the ground realities of the most vulnerable in the settlement may have been missed. Participants of the ToC highlighted challenges surrounding the ambiguity of ToC terminology. This challenge with ToC methodology has previously been highlighted (Stein and Valters, 2012) and suggests potential issues on how specific terms are understood. The ambiguity may change the way in which ToC is approached, and how the map is designed. Suggestions were made by participants in the current study to improve this by providing better simplifications of the constructs, and for these to be made available prior to the ToC workshops. Finally, while the individual interviews helped reflect on the use of ToC workshops in this context, having only five respondents was a further limitation to the findings.

### Conclusion

This study uses ToC to identify pathways influencing the implementation and maintenance of interventions addressing psychological distress and AUDs in a Ugandan humanitarian setting. Two pivotal pathways emerged: policy impact and mental health service delivery. The pathways, outcomes, interventions and assumptions outlined herein hold practical relevance for policy, research and practice. The participatory approach, involving multiple workshops and qualitative interviews, enhanced stakeholder ownership and commitment. It is important to note that the resulting ToC map is adaptable, pending assessment of intervention effectiveness, cost-effectiveness and reach. This study pioneers the use of ToC in planning sustainable interventions for AUDs among forcibly displaced persons in Uganda.

**Open peer review.** To view the open peer review materials for this article, please visit http://doi.org/10.1017/gmh.2023.93.

**Data availability statement.** Participants of this study did not agree for their data to be shared publicly, therefore supporting data are not available.

**Acknowledgments.** The authors express their gratitude to the participants who took part in this research study.

**Author contribution.** D.F. conceived and led the study as principal investigator. W.A.T. and E.K., as site-specific principal investigators, contributed to conception, study design and research activity. L.S.A. and A.M. led the ToC and qualitative analyses and played a significant role in article writing. D.T., the project coordinator, contributed to the research activity and provided written input. B.R., C.M. and A.N. as co-investigators made substantial contributions to the conceptualization, study design and writing. J.S. led the ToC workshops and provided written input. C.B., the research fellow, contributed to analysis, interpretation of findings and led the writing.

**Financial support.** This study has been funded by the National Institute for Health Research (NIHR) (using the UK's Official Development Assistance [ODA] Funding) and Wellcome (grant reference number 219468/Z/19/Z) under the NIHR-Wellcome Partnership for Global Health Research (Ref: HSRP496). The views expressed are those of the authors and not necessarily those of Wellcome, the NIHR or the Department of Health and Social Care.

**Competing interest.** The authors declare that they have no competing interests.

**Ethics statement.** This study has received ethical approval from the London School of Hygiene and Tropical Medicine Ethics Committee (Ref: 22729) and MildMay Uganda Research Ethics Committee (Ref: 0212-2020). All participants have provided written consent to participate, and for the publication of anonymized results in scientific journals.

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
