## [Reviewer Report]

Dear Prof. Chibanda and Prof. Bass,

We wish to submit an original article titled ‘Using Theory of Change to Plan for the Implementation of a Psychological Intervention Addressing Alcohol Use Disorder and Psychological Distress in Uganda’. We confirm that this work is original and has not been published elsewhere, nor is it currently under consideration for publication elsewhere. 

When developing complex interventions, it is recommended to investigate how context may influence intervention outcomes and implementation. In our paper we use a Theory of Change to identify pathways that influence the implementation and maintenance of a new transdiagnostic intervention (‘CHANGE’) aimed at addressing psychological distress and alcohol misuse in Uganda. We describe three workshops we conducted in Uganda to develop a Theory of Change map for psychological interventions in this context, and carried out semi structured interviews that were done with stakeholders to explore their experiences of participating in these workshops. Through the workshops, we were able to identify two pivotal pathways in the final Theory of Change map – a policy impact pathway and a mental health service delivery pathway. We consider the pathways, outcomes, interventions, and assumptions identified in this paper to be valuable for policy, research, and practice in the field of global mental health. As such, we believe that this is in line with the scope of this journal and would be very grateful for your consideration of this manuscript. 

Yours sincerely, 

Dr. Catharina Van der Boor

---

## [Reviewer Report]

The paper was interesting to read and describes a useful process which can inform others involved in adapting, testing and implementing complex interventions. Congrats to the authors on conducting and reporting this process.

Overall points

At first the paper seems to address psychological distress as well as AUDs, but as I read further, there were fewer and fewer mentions of the psychological distress component. Further, PM+ does not include alcohol use, except as an unhelpful coping strategy. What evidence-based intervention components were included to target alcohol use? Can you comment on this?

Also, much of the literature on brief interventions for alcohol use focuses on indicated prevention. Do you use diagnostic criteria to assess eligibility for the intervention? Or do you use a screening tool such as the ASSIST?

Has the ToC map been used to guide implementation? Is it being used for monitoring (are indicators being used)? Has it been adapted since the start of the project?

Minor points

1. Terminology: Different terms are used to refer to mental health conditions: mental disorders, mental health disorders. It would be a good idea to stick to one.

2. Line 53: “Kane” not capitalized in citation.

3. Line 274: “five start hotel” – this looks like a typo?

4. Line 392: in-text citation appears in 2 different formats

Impact statement

1. Lines 2-3: does the current study shed “light on the pressing issue of alcohol use disorders”, or the steps that stakeholders decided would lead them to the desired impact? The scope of the study seems to be overstated.

Background

1. The first paragraph is quite long; it would be good to break it up somewhat.

2. The topic sentence of the paragraph starting in line 66 does not seem to link well to the paragraph ideas. I’m assuming the paragraph is trying to say that ToC takes context into account. It sounds like the intervention has already been developed. What is the main reason for using ToC for this project? It would be good to highlight the known usefulness of ToC for this purpose. The De Silva paper mentions a range of uses but I suggest that you contextualise for your project; this would be easier for the reader to follow.

3. Line 76-77: “The aim of this study was to use ToC to summarize pathways that influence the planning for the implementation and maintenance of CHANGE for refugees in Uganda.” The words “that influence the planning” seem to be superfluous and are confusing. Do you mean to say that: “The aim of this study was to use ToC to summarize pathways to the implementation and maintenance of CHANGE for refugees in Uganda”?

4. Could you mention briefly why and how PM+ was chosen and adapted for CHANGE in Uganda? Has PM+ been tested in Uganda?

5. Can you also mention the intervention components for PM+ (eg problem solving and behavioural activation) and the alcohol use component?

Methodology

Setting

1. Line 96: I suggest rephrasing since “substance misuse” is not a non-communicable disease, but a health risk behaviour associated with various NCDs, including substance use disorders.

2. The rationale for the inclusion of Arua and the Rhino settlement is clear. For Kampala, is this because cases may be referred to Kampala, and senior Ministry of Health officials are based in Kampala?

3. It would be good to describe the current health services available in the Rhino settlement and any other psychosocial-related resources.

Participants

1. Can you clarify some of the titles used for participants? (block leader, women representative, refugee leader etc) It would be good to know if these people were refugees living in the camp or not. The limitations section suggests that refuges were not involved?

2. How were the subset of participants chosen for the semi-structured interviews?

Results

1. Much of the discussion focused on AUDs and not psychological distress. What was the reason for this? Psychological distress did not feature at all.

2. Lines 186-188: could you give ideas of the types of activities suggested for interventions 6 and 7?

Discussion & conclusion

1. Policy impact: the need for action and complementary policies across the government departments could have been discussed (regarding social determinants). Has there been any initiatives for health involving various departments? The COVID-19 pandemic spurred such action in many countries. (Ministries of Health began working with other ministries.) Is alcohol/substance use seen as a health issue or is the main role player here another government department?

2. Paragraph starting line 380: what relevant national mental health policies and plans are there in Uganda? It would be good to highlight local plans, knowledge and mechanisms in place to do this and also where the gaps lie (very broadly). You could also comment on the presence/absence of adequate substance use policy.

3. Lines 402 and 403: the study did not conduct a community survey. These lines seem to overstate the scope of the study.

4. Lines 389-415: These paragraphs could do with some restructuring. The discussion brings stigma in and then diverts to other statements around awareness and back to stigma. Lack of knowledge can be seen as part of stigma but the argument is a bit difficult to follow.

5. Interventions and stigma reduction interventions are mentioned but no sources are cited. Please review.

6. Lines 416-418: shift the focus of these lines to the recommendations by workshop participants that social determinants of mental health need to be prioritized. This statement in its current form seems to go beyond the scope of the study findings.

7. The rest of the paragraph can also be revised. The participants appreciated the collaborative effort which highlights the utility of ToC for mapping implementation. Yes, there is potential for sustainability and scale up if the process is repeated or the ToC is reviewed, but you don’t yet have any findings here since this is planning for implementation of a study. You can cite ToC success in health system strengthening and scale up phases elsewhere (see the work of Inge Petersen and colleagues – not just in PRIME).

8. You state that the process “enhanced stakeholder ownership and commitment” I would include the word “perceived” since surely this remains to be seen as the project unfolds?

---

## [Reviewer Report]

Using Theory of Change to Plan for the Implementation of a Psychological Intervention Addressing Alcohol Use Disorder and Psychological Distress in Uganda

14 November 2023

Thank you for the opportunity to review this paper. This paper reports on a participatory Theory of Change development process for AUD in conflict affected settings in Uganda. It reports the results of the Theory of Change (i.e. the map and describes the components) but also reflects on the process. They also report on five semi structured interviews with participants of the workshops.

Overall, this is a nice paper. I congratulate the authors on the nuance they have achieved in the results section by weaving together the assumptions and outcomes. I also commend the authors for including interviews of participants. The findings have resonated with my own experience running ToC workshops on the complexity of the ToC jargon.

I suggest some minor revisions

- Theory of Change purists will call Theory of Change an “approach” rather than a method because it relies on multiple methods for evaluation. I would also differentiate throughout that you used participatory Theory of Change workshops (rather than using a “Theory of Change” as there is no one universal approach to developing Theories of Change). ToC does not have to be participatory (although I think it should be!)

- In light of the above, in the methods could you be more specific about how your developed the ToC (rather than the ideal ToC components). I have found a table is often helpful for this. I am assuming that your ToC development did not only happen in the workshops but that there were multiple iterations after the workshops as well. It would be good to understand how this was done and how the information gained in the workshop was developed into a ToC. Some of this is covered in “procedure” so it may be worth consolidating this section with the “ToC workshops” section. Also, how were the audio recordings used. Were they transcribed or only used to check missing information?

- Prevalence is technically not a “rate” as there is no time component (unlike incidence which has a time component)

- The inclusion of stakeholder interviews is a really welcome addition to the ToC literature (very little is known about the experiences of stakeholders), however, a larger sample size would have produced more robust results.

- Pg 4,line 53 – Kane needs a capital letter

- Overall, I would have liked to understand how evidence was used to inform/modify/support the ToC map. At the moment, it’s really just the rationale.

ToC map

- Overall the ToC map is perhaps a little more like a process map than a theory of change. For example, it would be important to think about what changes as a result of the CHANGE intervention rather than just reporting that it had been offered.

- The outcome “Community is made …” could be reworded as “community is aware and….”

- The outcome “unemployment is addressed” could be rephrased as “Clients with alcohol problems are supported to find work/earn income”

- The interventions should be before the outcomes they produce, e.g. intervention 1 should come before the first outcome

---

## [Reviewer Report]

GENERAL COMMENTS

This paper describes the process and outcome of using a Theory of Change (TOC) to identify pathways influencing implementation and maintenance of the CHANGE intervention (an expanded version of PM+, which includes AUD management). In addition to sharing the primary outcome, a TOC map with two pathways - policy impact and mental health service delivery, the authors reflect on the utility of the TOC methodology and analyze findings from stakeholder interviews.

Theories of change are often developed for interventions or programs to guide the evaluation process. The concept of using a TOC to think about implementation and maintenance pathways prior to the full testing of an intervention is in line with “best practices” in scaling up as it seeks the early involvement and buy in of stakeholders in program design and helps to anticipate opportunities and barriers to implementation.

The TOC map does a good job of describing the contextual conditions necessary for sustained implementation and impact of the CHANGE intervention. One of the key strengths of the map is its visual simplicity. Often Theory of Change maps are overcomplicated, and their underlying messages become lost. This map strikes a good balance. Likewise, I appreciate the inclusion of the rational boxes/explanations, as it serves to ground the map in an existing evidence base.

SPECIFIC COMMENTS

I suggest the following minor modifications and clarifications:

1. The hyperlink to the CHANGE project is useful, but it would be helpful to include additional information about the intervention in the body of the article, including when the intervention is being implemented and tested, and by whom, as well as how this TOC intersects with the intervention evaluation work. Essentially, I would have appreciated understanding where the creation of this TOC fits within the larger picture of intervention development, testing, implementation etc. Perhaps a call out box, if the publisher permits it.

2. Please number or label the pink/orange boxes in the intermediate outcomes section. This will make discussing, reviewing and commenting on them much easier.

3. Please clarify which of the pink/orange boxes lie within the scope of the CHANGE intervention and which are outside of it (e.g. by circling or delineating the relevant boxes). For example, I’m not clear if assessment of clients with AUD is something that happens as part of the CHANGE intervention, or if CHANGE only refers to the enhanced PM+ treatment delivery in the next box.

4. Related to the comment above, it’s not clear what (or whom) the ceiling of accountability line refers to? Is everything to the right of the line assumed to be directly influenced by the CHANGE intervention? Or does this line simply differentiate the outcomes that will be monitored from those that will not? If so, who will be doing the monitoring?

5. Is the horizontal blue box at the bottom of the TOC an outcome or an assumption? If it’s an assumption, why is it not listed with the others on page 34?

6. It would be helpful if the authors outline their vision for using this TOC map moving forward (e.g. how will it be used within the life of the intervention, who might this be shared with, will it be treated as a living document etc).

7. Theories of Change for interventions are typically accompanied by indicators of the expected outcomes, which are then tracked over time as part of the evaluation process. I appreciate that this TOC is intended to provide a map of potential pathways to implementation and/or sustainability of the CHANGE intervention and thus such indicators may not be appropriate. However, I feel that the practical utility of this map lies in its potential as a tool for assessing the real-world viability of these pathways and identifying gaps/opportunities that could compromise/enhance the implementation and sustainability of the CHANGE intervention. Both program implementors and prospective funders interested in investing in the sustained implementation of the CHANGE intervention will want to understand the extent to which the contextual conditions required to support implementation currently exist or need to be created, and who could be engaged to do so. In other words, what are the chances of each “supporting” intervention and assumption actually being met? To this end, I encourage the authors to add detail to both the intervention and assumption boxes on page 34 that provides further information about the current implementation conditions (or at least how these will be assessed in future). For example:

Intervention: Train community health workers to screen for and identify AUDs in the refugee community and to select appropriate referral pathways (based on levels of severity and other associated problems).

Descriptive information: Do any such training programs already exist? What guidelines could be used? Are there currently any referral pathways? What are they and to where? Who is involved?

Intervention: Provide refugees in recovery from AUDs with vocational skills training

Descriptive Information: Do any vocational training programs currently exist in Rhino? Which agencies provide them/ could provide them (INGO, government)? Who needs to be engaged to make this happen?

Assumption: Mental health services are accessible to refugees.

Descriptive information: What accessible mental health services currently exist? Who provides them? Who uses them?

Assumption: Health care workers including social workers, community health care workers, and counsellors are available to provide services locally.

Descriptive information: How many of these workers are currently present in the camp or surrounding area? Who employs them (government, INGO)? What type of training have they received?

Assumption: Mental health plan supports the implementation of CHANGE.

Descriptive information: Is a mental health plan in place, at what level of gov? What opportunities exist to modify it in support of CHANGE? Who are the gatekeepers/key influencers?

8. The interviews about the Theory of Change process are an important addition to the paper, and capture both commonly held critiques about the process such as its complexity, inclusion issues (who should be involved in the process) and utility in a changing context. The final section does a good job of discussing the limitations of this process, but it could be beneficial if the authors reflect on why they did not engage a potential service user as part of the TOC process.

---

## [Reviewer Report]

Dear editor,

We kindly thank you and the reviewers for the helpful comments on the manuscript. We have addressed each of the comments provided.

Best wishes,

Catharina Van der Boor

---

## [Reviewer Report]

The revised manuscript does a good job of addressing reviewer comments and queries. The additional details regarding TOC process, origins of CHANGE intervention, mental health policy and access conditions in Uganda, and the proposed usage of the TOC within the context of intervention development are particularly helpful. The manuscript is a beneficial contribution to the existing literature.

Please note two suggested minor edits to the final text:

Line 18 – change 5 to “five”

Line 150 – delete the additional space between “the” and “draft”

---

## [Reviewer Report]

Thank you for addressing the points in detail and responding to my comments. One further comment: the sampling method for the interviews appears to be convenience sampling as the participants were not selected based on any particular characteristic, but rather on their availability. Congrats on your paper and valuable addition to the literature!